# The Mechanisms Underlying PTEN Loss in Human Tumors Suggest Potential Therapeutic Opportunities

**DOI:** 10.3390/biom9110713

**Published:** 2019-11-07

**Authors:** Hyeyoun Chang, Zhenying Cai, Thomas M. Roberts

**Affiliations:** 1Department of Cancer Biology, Dana-Farber Cancer Institute, Boston, MA 02215, USA; Hyeyoun_Chang@dfci.harvard.edu (H.C.); Zhenying_Cai@dfci.harvard.edu (Z.C.); 2Department of Biological Chemistry and Molecular Pharmacology, Harvard Medical School, Boston, MA 02215, USA; 3KIST-DFCI On-Site Lab, Department of Cancer Biology, Dana-Farber Cancer Institute, Boston, MA 02215, USA

**Keywords:** PI3K, PTEN, cancer

## Abstract

In this review, we will first briefly describe the diverse molecular mechanisms associated with PTEN loss of function in cancer. We will then proceed to discuss the molecular mechanisms linking PTEN loss to PI3K activation and demonstrate how these mechanisms suggest possible therapeutic approaches for patients with PTEN-null tumors.

The phosphoinositide 3-kinase (PI3K) signaling pathway regulates a wide range of vital cellular functions, including cell cycle progression, survival, motility, and metabolism [1]. Because of its central role in integrating multiple extracellular stimuli and intracellular processes, it is not surprising that PI3K signaling is one of the most frequently activated pathways in human cancers [2]. PI3Ks are lipid kinases that phosphorylate the 3’-hydroxyl group of phosphoinositides. The PI3K family is categorized into classes I, II, and III based on structure and substrate specificity. Thus far, only class I PI3Ks have been shown to be directly linked to tumorigenesis and cancer development. Class I PI3Ks are heterodimers composed of a p110 catalytic subunit (p110α, p110β, p110γ, or p110δ) and a regulatory subunit (p85 or p101), and can be directly activated by cell surface receptors. They are further divided into subclasses IA and IB based on the specific regulatory subunit that couple them to receptors. Both p110α and p110β are commonly expressed in cells outside the immune system and are often activated in solid tumors, while p110δ and p110γ are expressed in cells of the immune system and are often activated in hematological malignancies [3,4,5]. The main negative regulator of this pathway is phosphatase and tensin homolog (PTEN) [6,7,8]. PTEN dephosphorylates phosphatidylinositol (3,4,5)-triphosphate (PIP_3_), a lipid second messenger produced following PI3K activation, to phosphatidylinositol (4,5)-bisphosphate (PIP_2_), thereby antagonizing the PI3K signaling pathway. PTEN also functions as a protein phosphatase in the cytoplasm, where it has been shown to inhibit cell migration and induce cell cycle arrest [9,10], or in a phosphatase-independent manner in the nucleus to regulate chromosome stability, DNA repair, and apoptosis [11,12,13,14]. Analogously to direct oncogenic activation of PI3K proteins or upstream activators, loss-of-function mutations or epigenetic silencing of the *PTEN* tumor suppressor has been shown to drive the development of a variety of human cancers [15].

## 1. PI3K Mutations in Human Cancers

Hyperactivation of the PI3K pathway is frequently found in many types of human cancer. Increased activity of the PI3K pathway is often associated with tumor progression and resistance to cancer therapies [16]. Enzymatic activation of the PI3K pathway occurs most frequently through activating mutations in *PIK3CA,* the gene encoding p110α, or through gain-of-function mutations/amplification in receptor tyrosine kinases (RTKs) and oncogenes such as *KRAS*. Preclinical studies have shed light on the reasons that *PIK3CA*, but not *PIK3CB*, the gene encoding p110β, is so frequently mutated in human tumors. In genetically-engineered mouse (GEM) models, the activation of p110α, either directly via activating mutations at so-called hotspots due to their high mutational frequency (E545K and H1047R), or indirectly via activating mutations or overexpression of RTKs, such as Her2, is sufficient to drive tumor formation. In the case of tumors driven by activated RTKs, the deletion of p110α is sufficient to block tumor initiation, while deleting p110β can actually increase tumor size [17]. This appears to occur in part because both enzymes compete for activation motifs on the RTKs, and p110α has a much higher specific activity than p110β. This high specific activity may also explain why the *PIK3CA* gene is a better mutational target in human tumors, though other mechanisms may be relevant. However, the quantitation of class IA PI3Ks using quantitative mass spectrometry revealed at least five times higher loading of p110α-heterodimers than p110β-heterodimers to activated platelet-derived growth factor receptors in mouse embryonic fibroblasts [18]. The other major source of PI3K pathway activation in tumors is the loss of the tumor suppressor PTEN. Notably, as will be discussed in more detail below, the activity of p110β, the weaker of the two enzymes in terms of catalytic activity, is often essential in PTEN-deficient tumors [19,20].

An integrative analysis of cancer genomics using cBioPortal reported that *PIK3CA*, *PTEN*, and/or *AKT1* mutations are present in over 40% of uterine, lung, breast, prostate, brain, and head and neck cancers [21,22]. A targeted, massively parallel sequencing project that assessed the frequency of mutations in 47 genes, including *PIK3CA*, *PTEN*, and *AKT1*, in 19,784 consecutive diverse solid tumors from 60 countries supported previous studies that aberrations in the PI3K pathway are frequently observed in a broad range of human tumors [23]. In addition to uterine, breast, prostate, and lung cancers, the study found that anal, liver, and colorectal cancers also contained mutations in PI3K pathway genes with high frequency.

## 2. Prevalence of PTEN Mutations in Cancer

As one of the most frequently mutated tumor suppressor genes in human cancer, *PTEN* is altered in a wide range of human cancers, including breast cancer, prostate cancer, endometrial cancer, and glioblastoma [24,25]. *PTEN* was first discovered as a frequently mutated tumor suppressor gene on chromosome 10q23 [6,8]. Mutation rates in at least one allele of this locus are estimated to occur in the range of 50%–80% in multiple sporadic tumors, including endometrial carcinoma, glioblastoma, and prostate cancer, and at 30%–50% in breast, colon, and lung tumors [26]. A cancer genomics study revealed that the tumor suppressor locus in the human genome with the greatest selection for homozygous deletion is *PTEN* [27]. Somatic mutations on *PTEN* are scattered along the entire gene. There are mutational hotspots (R130, R173, R233, K267, and N323), but mutations at these hotspots have not been shown to be specific for a certain type of cancer [28]. R130 and R173, as well as the majority of the missense mutations found in human tumors, are located in the phosphatase domain, affecting the phosphatase activity of PTEN [29]. R233, K267, and N323 are found in the C2 domain, which is essential for lipid binding, thus reducing the ability of PTEN to associate with the plasma membrane [30]. Partial loss of *PTEN* function is frequently observed in the clinic. Hemizygous loss of *PTEN* leads to cancer progression in the prostate, thyroid, colon, lymphatic system, mammary gland, and endometrium [31,32]. Studies in mice have shown that even a small decrease in PTEN expression is sufficient to promote cancer susceptibility [33,34,35]. Although the complete loss of *PTEN* is observed more frequently in endometrial carcinoma and glioblastoma, it appears in less than 15% of prostate, breast, colon, and lung cancers. Even in the cancers with the highest biallelic mutation frequencies, monoallelic mutation and loss of protein expression are observed more commonly [15,36,37]. Germline mutations of *PTEN* are also associated with genetic disorders with an increased risk of cancer, collectively referred to as PTEN hamartoma syndromes (PHTS), which include Cowden syndrome, Bannayan–Zonana syndrome, Lhermitte–Duclos syndrome, Proteus syndrome, and Proteus-like syndrome [38]. The majority of germline mutations on *PTEN* are nonsense or frame-shift mutations and occur in the C-terminal region, resulting in premature stop codons and truncations of the protein in approximately 70% of the total mutations studied [38]. Experiments performed in GEM models have demonstrated that mice with tissue-specific *Pten*-knockout develop tumors in various organs, supporting the role of *PTEN* as a potent tumor suppressor [31,39,40].

## 3. Epigenetic, Transcriptional, and Post-Transcriptional Regulation of PTEN

In addition to the frequent occurrence of PTEN mutations in various cancer types, PTEN levels are also downregulated at both transcriptional and post-transcriptional levels.

### 3.1. Transcriptional Regulation of PTEN

*PTEN* mRNA levels are regulated by several transcription factors. The p53 tumor suppressor upregulates PTEN expression in response to irradiation and other sources of DNA damage via direct binding to a cis-element in the PTEN promoter to trans-activate PTEN. The induction of p53 in cells elevates PTEN at both mRNA and protein levels in a variety of cell lines [41]. Loss of p53 caused a decrease in PTEN expression in mouse embryonic fibroblast (MEF) cells [42]. EGR1 upregulates *Pten* transcription by direct binding to a GCGGCGGCG site in the 5’ untranslated region [43]. It has been reported that EGR1 levels predict *PTEN* expression and survival in non-small-cell lung cancer patients [44]. The EGR1-PTEN axis is correlated with tumor sensitivity to radiation, cell growth, invasion, and epithelial-mesenchymal transition (EMT) in various cancers [44,45,46,47,48,49,50]. Notch-1 signaling regulates the transcription of *PTEN* through the dual-function transcription factor CBF-1. In the absence of active Notch-1 signaling, CBF-1 binds to the *PTEN* promoter and represses *PTEN* expression. Overexpression of constitutively active Notch-1 converts CBF-1 into a transcriptional activator and induces PTEN expression in 293T cells and human prostate cancer cells [51,52,53]. Defects in Notch-1 signaling may contribute to the loss of PTEN expression in prostate adenocarcinoma [53]. PPARγ can bind two response elements located 23.3 kb upstream of PTEN and upregulate PTEN expression, thereby modulating allergic inflammation and tumor suppressor functions in inflammatory cells and some cancer cells [54,55,56,57]. The regulation of PTEN expression by NF-κB is not clearly delineated. The p50 and p65 subunits of NF-κB suppress PTEN expression through either direct or indirect mechanisms, while the p100 subunit stabilizes PTEN mRNA via inhibition of miR-494 [58,59,60,61]. JNK signaling can either induce or repress PTEN expression through its action on different downstream transcriptional factors, ATF-2 and c-Jun [62,63,64]. Previous research also suggests Bmi-1 and Snail1 repress PTEN expression [65,66].

### 3.2. Epigenetic Regulation of PTEN Expression

Aberrant methylation of the PTEN promoter provides another mechanism for the loss of PTEN expression. Hypermethylation of CpG islands in the PTEN promoter is found in various types of cancers, including breast, lung, ovarian, cervical, colorectal, and thyroid cancer, as well as melanoma, gastric carcinoma, and metabolic syndrome [67,68,69,70,71,72,73,74,75,76,77,78]. In addition, SALL4 recruits the Mi-2/Nucleosome Remodeling and Deacetylase (NuRD) complex to the promoter region of PTEN, and represses PTEN transcription via the complex’s histone deacetylase activity [79,80]. The blocking of SALL4 in glioma can alter PTEN expression and PI3K signaling [80]. Evi1 interacts with several polycomb complex proteins to form a repressive complex for PTEN downregulation, providing a novel epigenetic form of regulation of the PI3K pathway in leukemia [81].

### 3.3. miRNA and PTEN Expression

A number of miRNAs have been shown to participate in the regulation of biological processes by targeting *PTEN*. For example, *Mir-21* binds to the 3’-UTR of *PTEN* mRNA and inhibits *PTEN* expression [82]. *MiR-21* confers resistance to some chemotherapy drugs and radiation therapy in cancers [83,84,85,86]. This effect appears to be mediated via modulation of PTEN since downregulation of PTEN in the HCC cell attenuates the effects of anti-miR-21 on growth and invasion [82]. Similarly, application of the PI3K inhibitor LY294002 in gastric cancer cell line SGC7901 could abrogate miR-21-induced cell survival and resistance to chemotherapeutic agent cisplatin [83]. *MiR-29* regulates *PTEN* in various ways. Previous research demonstrates that YAP1 downregulates *PTEN* by inducing *MiR-29* to inhibit PTEN translation. In this case, PTEN knockdown mimics the YAP-induced increase in cell size and LY294002 treatment rescues the effect of YAP on cell size [87]. *MiR-29a* may confer the acquisition of resistance to adriamycin in breast cancer cells by the downregulation of *PTEN* and activation of the PI3K pathway [88]. The effect of *MiR-29* on PTEN level is a cell type-dependent. Studies in hepatic stellate cells and liver fibrosis models suggest that *MiR-29* upregulates *PTEN* expression by targeting DNA methyltransferases such as *Dnmt1*, *Dnmt3b*, and *Set1a* to prevent excess liver fibrosis [89,90]. Finally, the *MiR-130* family downregulates *PTEN* to promote cell proliferation, invasion, migration, and EMT in cancer [91,92,93].

### 3.4. Long Non-Coding RNA and PTEN Expression

Long non-coding RNAs can upregulate *PTEN* expression either by functioning as a decoy for *PTEN*-targeting miRNAs or by altering *PTEN* promoter methylation. *PTENP1* is a processed pseudogene with high homology to *PTEN*. *PTENP1* is targeted by *PTEN*-targeting miRNAs, and can function as a competing endogenous RNA (ceRNA) to protect *PTEN* from miRNA recognition [94]. The stabilization of *PTEN* by *PTENP1* inhibits cell proliferation, invasion, and migration in cancers [95,96,97,98,99,100,101]. Other long non-coding RNAs, including *GAS5*, *FER1L4*, *XIST*, *MEG3*, and *NBAT,* have also been reported to sequester miRNAs that specifically target *PTEN*, thereby stabilizing PTEN mRNA and modulating PTEN function [102,103,104,105,106,107,108,109,110,111,112,113,114,115,116]. The long non-coding RNA *HOTAIR* regulates *PTEN* expression via several different mechanisms [117,118,119,120]. Depletion of *HOTAIR* in laryngeal squamous cell carcinoma decreases *PTEN* methylation to reduce invasion and tumor growth [117]. *HOTAIR* can also upregulate *PTEN* expression by targeting *miR-19*, *miR-29b*, and *miR-17-5-p* [118,119,120].

## 4. Post-Translational Modifications of PTEN

The activity, stability, and subcellular localization of PTEN can be regulated by several post-translational modifications (Table 1). The sections below will briefly discuss them.

### 4.1. Phosphorylation

The phosphorylation of PTEN has different functions on the activity, stability, and subcellular localization. CK2 phosphorylates several Ser (S)/Thr (T) residues in the C-terminal tail of PTEN to decrease the activity and increase the stability of PTEN [121,122]. The mutation of two CK2 phosphorylation sites, S370 and S385, to alanine, accelerates proteasome-mediated degradation of PTEN and partial proteolysis by caspase-3 [122,123]. PTEN is also phosphorylated on T366 by GSK3β. Blocking the phosphorylation of T366 by either mutation or GSK3 inhibition in glioblastoma cell lines led to a stabilization of the PTEN protein [124]. Dephosphorylation at T382 and T383 leads to an increase in the PTEN degradation rate and enhanced binding affinity to the PDZ2 domain of MAGI-2 [125]. *H. pylori* inactivates PTEN by inducing PTEN phosphorylation at residues S380/T382/T383, resulting in the enhanced survival of gastric epithelial cells [126]. The mutation of four phosphorylation sites in the C-terminal tail (S380, T382, T383, and S385) to alanine increases the membrane association of PTEN [127,128]. The phosphorylation of tyrosine 240 is frequently found in glioblastoma (GBM) patient samples [130,131]. A recent study suggested that ionizing radiation treatment induces Y240 phosphorylatoin of PTEN by FGFR2, which facilitates nuclear localization and chromatin binding of PTEN [130]. Y240 phosphorylation confers resistance to EGFR inhibition and ionizing radiation both in vitro and in vivo, and is associated with decreased overall survival in patients. Both fibroblast growth factor receptors and SRC family kinases mediate PTEN Y240 phosphorylation, which functions independently of the lipid phosphatase activity of PTEN [130,131].

### 4.2. Ubiquitination

The activity and stability of PTEN has been reported to be down-regulated by monoubiquitination and polyubiquitination. Two main monoubiquitination sites, K13 and K289, are important for PTEN nuclear localization and stability [129]. Nedd4-1 has been reported to ubiquitinate PTEN, thus regulating its stability and nuclear localization [129,132,133]. Another study found that Nedd4-1 is dispensable for the regulation of PTEN stability and localization in MEF and 293T cells, suggesting the regulation may act in a cell type and tissue specific manner [148]. The regulation of PTEN mono- and polyubiquitination involves multiple ubiquitin ligases. CHIP and RNF-146 are two additional candidate E3 ligases for PTEN [134,135]. WWP2 physically interacts with PTEN and mediates its degradation through a ubiquitination-dependent pathway [136,137]. PTEN protein levels are also elevated in *WWP1-*depleted cells [138]. WWP1 mediates the polyubiquitination of PTEN at K27, thus suppressing the dimerization, membrane recruitment, and function of PTEN [138,139]. XIAP also promotes PTEN ubiquitination and degradation. Upregulation of XIAP by TGF-β3 causes a reduction of PTEN in both the cytosol and the nucleus [140].

### 4.3. Oxidation

The Cys124 residue of PTEN forms a disulfide bond with Cys71 in response to oxidation by both endogenous and exogenous hydrogen peroxide, leading to the reversible inactivation of the lipid phosphatase function of PTEN [141,142]. In pancreatic cancer cells expressing COX-2 or 5-LOX, PTEN is oxidized and inactivated via arachidonic acid (AA) metabolism [149]. Several proteins are thought to be involved in the regulation of PTEN oxidation. Txnip ablation increases PTEN oxidation by thioredoxin NADP(H) reduction [150]. AIF directly interacts with PTEN and prevents oxidative inactivation [151].

### 4.4. Acetylation

PCAF mediates acetylation of PTEN at lysine 125 and 128 to inhibit its activity in the regulation of the PI3K/AKT pathway [143]. The acetylation of PTEN at lysine 402 by CBP facilitates its interaction with PDZ domain-containing proteins [144]. Application of two non-selective HDAC inhibitors, TSA and suberoylanilide hydroxamic acid (SAHA), induced PTEN membrane translocation through PTEN acetylation at K163 by inhibiting HDAC6 [145].

### 4.5. S-Nitrosylation

Nitric oxide regulates the activity of PTEN via S-nitrosylation during transient global ischemia in the rat hippocampus [152]. The S-nitrosylation of PTEN at C83 inhibits the enzymatic activity of PTEN [146]. *PARK2* depletion contributes to AMPK-mediated activation of endothelial nitric oxide synthase (eNOS), promoting both S-nitrosylation and ubiquitination of PTEN [153,154]. NOS1 prevents excessive autophagy and promotes the survival of nasopharyngeal carcinoma cells through the S-nitrosylation of PTEN [155].

### 4.6. Ribosylation

TNKS1 and TNKS2 interact with and ribosylate PTEN, which promotes the recognition of PTEN by RNF146, leading to PTEN ubiquitination and degradation [135].

### 4.7. Sumoylation

The SUMO1 modification of PTEN at Lys266 in its C2 domain was shown to increase the localization of PTEN at the plasma membrane, resulting in the suppression of the PI3K-AKT pathway signaling and inhibition of anchorage-independent cell growth and tumor growth in vivo [147].

## 5. Mechanisms of PI3K Activation in PTEN-Null Tumor Cells

The loss of PTEN in a tumor amounts to removing the brake on PI3K signaling (Figure 1). However, exactly which PI3K isoforms and mechanisms are involved in the activation of the PI3K pathway in PTEN-null cells can vary from tumor to tumor. Thus, in preclinical models of tumors featuring PTEN loss accompanied by expression of oncogenes or activated RTKs, p110α was found to be the major source of PI3K signaling [156]. Due to p110α’s high specific activity and specific binding to Ras proteins, the result is hardly surprising [157]. However, in GEM tumor models driven by the knockout of PTEN and in a number of PTEN-null human tumor cell lines lacking expression of oncogenes or activated RTKs, p110β is often the major source of PI3K signaling [19,20,158]. It is remarkable that the less active isoform is generating the PI3K signal in response to PTEN loss in these models.

To address this question, Cizmecioglu et al. first examined how p110β is activated under physiological conditions by GPCRs [159]. Since the plasma membrane is the major site of PI3K activation and integration of divergent growth factor signals, the authors initially aimed to understand how spatial compartmentalization in the plasma membrane might contribute to the functions of the ubiquitous class IA PI3K isoforms, p110α and p110β. GPCRs, which are known to signal via p110β, are localized in lipid rafts. Cizmecioglu et al. reported that binding of Rac1, a known binding protein of p110β, was necessary to localize p110β to membrane rafts, which increases the likelihood of its activation by GPCR. On the other hand, p110α, which is not associated with GPCR signaling, was found in non-raft regions of the plasma membrane. Interestingly, when p110α was artificially tagged to direct it to rafts, GPCRs were able to mediate the activation of AKT via p110α. It had also been previously shown that upon PIP_3_ production, raft nanodomains were involved in recruiting AKT to the plasma membrane [160]. Cizmecioglu et al. further demonstrated that p110β-dependent PTEN-null tumor cells critically rely on raft-associated PI3K signaling [159]. A unique positive feedback loop was suggested in which p110β is localized for activation by Rac [161] and Rac in turn is activated by PIP_3_ via PIP_3_-activated guanine-nucleotide exchange factors (GEFs) [162,163]. In other words, p110β and Rac form a positive feedback loop to generate the key signals associated with Pten loss that leads to cell proliferation and migration (Figure 1).

The data on the Rac/p110β feedback loop described above (see [161] for details) leave open a big question: How is the feedback loop activated in the first place? Or, more specifically, what is initially activating p110β in PTEN-null tumors? While in theory this could be achieved via interactions with Gβγ subunits of GPCRs, as is the case physiologically, previous data suggests this is often not true for PTEN-null tumors [159]. One clue to the identity of the mechanism providing the “foot on the accelerator” of the PI3K pathway was found by a proteomic screen for new p110β-interacting proteins. CRKL was identified as a protein preferentially associated with p110β [164]. Mechanistic studies by Zhang et al. revealed that, in addition to p110β, CRKL binds to p130Cas, which is, in turn, activated via phosphorylation by Src [165,166]. The depletion of CRKL was found to downregulate p110β-dependent PI3K signaling in PTEN-deficient cancer cells. Zhang et al. suggested that a PTEN/FAK/Src/p130Cas axis may activate CRKL/p110β [164]. The cooperation of Src inhibition with p110β inhibition to suppress the growth of PTEN-null cells was also demonstrated. Significantly, the data indicate a potential mechanism behind the link between PTEN loss and p110β activation.

Additionally, there may be an interesting feedback loop connecting YAP activation, *PTEN* loss, and p110β activation. As previously mentioned, YAP activation decreases PTEN expression via miR-29 [87]. Activation of YAP also directly increases p110β expression through TEAD [167]. Reciprocally, p110β activation via *PTEN* loss is known to activate Rac, which in turn can activate YAP [159,168]. Thus, there exists a putative positive feedback loop in which p110β activates YAP via Rac and, once activated, YAP in turn increases p110β expression and potential activity. Notably, a signature of YAP activation is seen in the set of PTEN mutant cell lines which are sensitive to the p110β inhibitor KIN-193 [169].

## 6. Turning Mechanistic Insights into Possible Therapies

Early clinical trials with so-called pan-PI3K inhibitors were unimpressive for most PTEN-null tumor indications [170,171]. This is likely because early pan-inhibitors such as BKM120 and GDC-0941 were roughly 10–20 times more potent on p110α than p110β. However, recent trials have begun using p110β-targeted compounds such as AZD8186 (NCT01884285) and GSK2636771 (NCT01458067), and early results are more promising [172,173]. An AKT inhibitor from Genentech, GDC-0068, has also had promising early results in PTEN-null tumors [174].

Inevitably, single agent therapies are likely to have only short-lasting effects in solid tumors. Hence, preclinical research is now focused on finding combination therapies, largely building on p110β inhibitors. Unlike compounds that target p110α, which have shown high on-target toxicity due to effects on insulin signaling, a pure p110β inhibitor is expected to have relatively low on-target adverse effects. Indeed a mouse with whole-body knock-in of a kinase-dead allele of p110β is fairly healthy with only relatively minor insulin resistance [175]. One example of a combination therapy studied is the combination of a p110β inhibitor and a Src inhibitor targeting the signaling cascade of Src/p130Cas/CRKL/p110β. The results showed suppressed growth of PTEN-null human cancer cells [164]. Another example is a study which found that PTEN loss suppresses androgen receptor transcription factor activity and androgen-responsive gene expression, thereby promoting castration-resistant prostate cancer (CRPC) development [176]. Based on the pre-clinical study, androgen deprivation therapy accompanied by a p110β inhibitor may prove to be beneficial in treating CRPC. In fact, there is an ongoing clinical trial (NCT02215096) that combines a p110β inhibitor with an antiandrogen in treatment of metastatic CRPC [177].

### 6.1. Combining p110β Inhbition and Immunotherapies

Recently, exciting new roles of PTEN in immune regulation and inflammatory diseases have been discovered. PTEN has been found to play important roles in both T cells and in cloaking tumors from the immune system. A study in mice demonstrated that the T cell-specific loss of PTEN in mice causes defects in central and peripheral tolerance and increase susceptibility to T cell lymphoma and leukemia [178]. More exciting in terms of therapeutic implications is the tumor cell-specific role of PTEN in the tumors’ interaction with the immune system. In a recent paper, the loss of PTEN in patients was found to be correlated with reduced T cell infiltration at tumor sites and a poorer response to PD-1 blockade therapy [179]. In this study, Peng et al. showed that the loss of PTEN in tumor cells boosted the level of immunosuppressive cytokines, creating a microenvironment that allows tumor cells to escape from T cell-mediated cell death. In a follow-up pre-clinical study, the group demonstrated that a combination of an OX40 agonist antibody and a p110β inhibitor induce robust and durable anti-tumor T cell immunity in melanoma mouse models [180]. Furthermore, a clinical study (NCT03131908) is in progress that combines a p110β inhibitor with an anti-PD-1 antibody in patients with metastatic melanoma and PTEN loss [181]. Thus, it is likely that the most effective treatment for PTEN null tumors will involve a combination of targeted therapies, such as p110β inhibitors and immunotherapies.

### 6.2. PTEN-Deficient Synthetic Lethality 

Synthetic lethality is said to occur for two genes when alteration in either gene alone does not affect cell viability but perturbation of both genes at the same time leads to cell death. The concept of synthetic lethality can be applied to cancer therapy by inhibiting one protein activity in cancer cell harboring a loss of function mutation of the second gene to induce cell death [182]. Numerous examples of synthetic lethality have been reported for PTEN. Homologous recombination is deficient in PTEN null human tumor cells, which sensitizes the cells to poly ADP-ribose polymerase (PARP) inhibitors [183]. Similarly, knockdown of *CHD1* inhibits cell growth in PTEN-null prostate cancer cells while it has minimal effect in PTEN-intact cell lines [184]. RAD51 is a key regulator involved in DNA double strand break repair and homology-directed repair. Treatment of PTEN-deficient glioma cells with a cell-penetrating antibody against RAD51, 3E10, leads to an accumulation of DNA damage causing decreased proliferation and increased cell death compared to isogenic PTEN-proficient controls [185]. ITGA5 knockdown and the pharmacologic inhibition of BCL2-BCLX_L_ had a synergistic effect with PI3K/AKT inhibitors in PTEN-mutant prostate cancer cells, such as LNCap and PC3, but not in PTEN-proficient DU145 cells [186]. The ATM kinase inhibitor, KU-60019, preferentially sensitizes PTEN-deficient MDA-MB-468 breast cancer cells to cisplatin, while only slightly enhancing sensitivity of PTEN-wild type breast cancer cells [187]. BRG1 was identified in a CRISPR-Cas9-based screen in 22RV-1 cells for epigenetic regulators specifically required in PTEN-deficient PCa cells. The BRG1 antagonist, PFI-3, inhibits PTEN-deficient tumor progression [188]. In PTEN-null GBM models, LOX inhibition markedly suppresses macrophage infiltration and tumor progression [189]. Finally, the metabolic regulator, pyruvate dehydrogenase kinase 1, was identified as a synthetic-essential gene and potential therapeutic target in PTEN-deficient cancers [190].

### 6.3. Reactivation of PTEN

Recent research suggested that it is possible to reactivate PTEN expression/activity for cancer treatment [48,139,191]. Thus, HDAC inhibition induces increased PTEN levels in an EGR1-dependent manner, promoting apoptosis in synovial sarcoma [48]. Oxidized PTEN is preferentially reactivated by the thioredoxin system. This PTEN reactivation was inhibited by inducible depletion of thioredoxin 1 in intact living cells [191]. In a recent paper, researchers found that indole-3-carbinol (I3C) was able to inhibit WWP1-mediated PTEN K27-linked polyubiquitination both both in vitro and in vivo. Since WWP1 expression is regulated by MYC, I3C treatment may be especially effective in tumors driven by MYC [139].

## 7. Conclusions and Perspective

Tumors lacking the PTEN expression comprise one of the largest tumor classes. Despite the fact that PTEN loss results in abnormally high PI3K pathway activation, therapeutic approaches to this tumor class have been disappointing. Historically, the development of anti-cancer therapeutics targeting tumor suppressors has been particularly challenging. In the case of PTEN, this challenge has been exacerbated by the multiple functions of the PTEN protein and its highly complex modes of regulation. However, with the advent of p110β-selective PI3K inhibitors and promising combinations of these inhibitors with both targeted and immunotherapies, it can be hoped for that a more favorable outlook for patients with PTEN-null tumors is on the horizon.

## Figures and Tables

**Figure 1 biomolecules-09-00713-f001:**
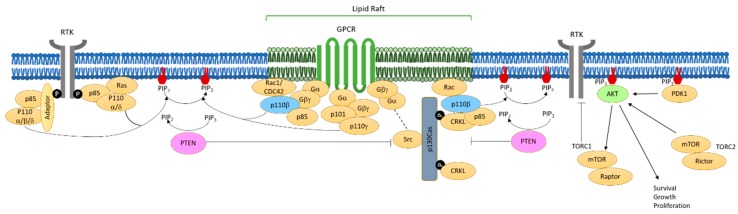
Overview of PI3K-AKT signaling pathway. Class I PI3Ks are heterodimers composed of a catalytic subunit (p110) and a regulatory subunit (p85 or p101). Once activated by cell surface receptors, PI3Ks phosphorylate PIP_2_ to PIP_3_, which activates AKT for cell survival, growth, and proliferation. PTEN is the main negative regulator of the signaling pathway. PTEN-null tumor cells often rely on p110b for PI3K signaling, and the figure above depicts the potential mechanism of p110b activation via CRKL/p130Cas/Src/Rac, which results in their localization in lipid rafts in the absence of PTEN.

**Table 1 biomolecules-09-00713-t001:** Post-translational modifications of PTEN.

Modification	Modifier	Site
**phosphorylation**	CK2 [121,122,123]	S370, S385 [121,122,123]
	GSK3β [124]	T366 [124]
		T382, T383 [125] S380, T382, T383 [126] S380, T382, T383, T385 [127,128]
	FGFR,SRC family kinases [129] FGFR2 [130]	Y240 [130,131]
**Ubiquitination**	Need4-1 [129,132,133] CHIP [134] NRF146 [135] WWP2 [136,137] WWP1 [138,139] XIAP [140]	K13, K289 K48 [134] K27 [139]
**Oxidation**	ROS [141,142] Thioreductase [142]	C71, C124 [141,142]
**Acetylation**	PCAF [143]	K125, K128 [143]
	CBP [144]	K402 [144]
	HDAC6 [145]	K163 [145]
**S-nitrosylation**	NO [146]	C83 [146]
**Ribosylation**	TNKS1, TNKS2 [135]	
**Sumoylation**	SUMO1 [147]	K266 [147]

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
