# Peer review of "The Mechanisms Underlying PTEN Loss in Human Tumors Suggest Potential Therapeutic Opportunities"

_biomolecules, 2019, doi:10.3390/biom9110713_

Round 1

Reviewer 1 Report

This concise review of PTEN loss in cancer will be of considerable interest to readers. Overall, the review is accessible, interesting, and clear. It is well-written. A few suggestions are made to enhance the impact of the review:

The title focuses on targeting tumors with PTEN loss, but relatively little focus is placed within the text on the results of targeting, particularly the problems of bypass signaling, feedback activation, and pharmacokinetic issues that have limited the efficacy of AKT inhibitors, and other inhibitors that block the pathway. Rather, the focus is largely on the biology, which is illuminating. Perhaps modifying the title to more accurately reflect the emphasis of the review would be of value? Alternatively, adding more information about what has happened in the clinic, and what has been learned with regard to therapeutic resistance could be included. I favor the focus on the biology, as it is more in the spirit of how this review was written. However, a few statements regarding the therapeutic challenges would be of value. It is probably worth discussing the growing literature on targeting PTEN-deficient synthetic lethal vulnerabilities as a therapeutic strategy. Line 75 discusses PTEN mutational hot spots. Readers would benefit from some text explaining the impact of these mutations, their domains, and the impact on enzymatic function. There are some recent data about tyrosine 240 phosphorylation of PTEN and its impact in cancer, that would be beneficial to briefly mention.

Reviewer 2 Report

Chang, Cai and Roberts provide a review about PTEN biology and approaches to target tumors with PTEN loss of function. Although this topic has been reviewed before, the field continues to evolve rapidly. The authors have provided a thorough and authoritative review that highlights important recent findings, while also capturing some literature that might be underappreciated.

The middle section of the review provides an exhaustive survey of papers that report epigenetic and post-translational regulation of PTEN expression and function. However, the authors do not return to this subject in the section on targeting PTEN-deficient tumors. Instead they focus almost entirely on PI3K inhibitors and combinations with other kinase inhibitors or immunotherapies. The review would benefit if the authors could elaborate on approaches to re-activate PTEN in cells that have intact PTEN genes. For example, recent work from the Pandolfi lab (ref. 135) provides a novel avenue for therapeutic intervention via inhibitors of WWP1. They might also discuss the possibility that epigenetic therapies might work in part by un-silencing PTEN transcription.

Parts of the review should be revised to better engage and inform the reader. For example, some paragraphs have topic sentences with directionless verbs (modulate, regulate):

“Long non-coding RNAs can modulate PTEN expression”

“CK2 phosphorylates several Ser (S) /Thr (T) residues in the C-terminal tail of PTEN to regulate its activity and stability”

“PTEN has been reported to be regulated by both monoubiquitination and polyubiquitination”

A related concerns is that some of the paragraphs present lists of papers with relatively few “big picture” insights or models. Overall, the section on post-translational modifications would benefit from a summary and perhaps a figure or table. In the section on miRNAs and lncRNAs, it would be helpful to describe what experimental approaches have been done to establish the primary importance of PTEN as a target of regulation (since these RNAs have multiple targets).

Other comments:

In section 1, near lines 50-52, the authors should consider citing and discussing work from Stephens and Hawkins that used knock-in tags and quantitative mass spectrometry to show distinct recruitment of p110alpha versus p110beta to RTKs (PubMed ID 30442661)

In section 3.3, is there any way to resolve the opposing findings about MiR-29?

Some of the references seem incorrect: 162, 163, 174, 178

On line 260, I believe the correct Genentech compound is GDC-0941, not GDC-0049.

On lines 286-287, can the authors elaborate on how loss of PTEN creates “a microenvironment that allows tumors to evade immune suppression.”
